# A Transmembrane Histidine Kinase Functions as a pH Sensor

**DOI:** 10.3390/biom10081183

**Published:** 2020-08-14

**Authors:** Ana Bortolotti, Daniela Belén Vazquez, Juan Cruz Almada, Maria Eugenia Inda, Salvador Iván Drusin, Juan Manuel Villalba, Diego M. Moreno, Jean Marie Ruysschaert, Larisa Estefania Cybulski

**Affiliations:** 1Departamento de Microbiología, Facultad de Ciencias Bioquímicas y Farmacéuticas, Universidad Nacional de Rosario-Argentine National Research Council—CONICET, Suipacha 531 CP 2000, Argentina; anabortolotti@gmail.com (A.B.); daniela.vazquez@live.com (D.B.V.); almadajuancruz@gmail.com (J.C.A.); indamariaeugenia@hotmail.com (M.E.I.); jmvilla_ls@hotmail.com (J.M.V.); 2Departamento de Químico-Física, Facultad de Ciencias Bioquímicas y Farmacéuticas, Área Física, Universidad Nacional de Rosario, Suipacha 531, Rosario S2002LRK, Santa Fe, Argentina; sdrusin@fbioyf.unr.edu.ar; 3Instituto de Química de Rosario (CONICET-UNR), Suipacha 570, S2002LRK Rosario, Santa Fe, Argentina; moreno@iquir-conicet.gov.ar; 4Área Química General e Inorgánica, Departamento de Química-Física, Facultad de Ciencias Bioquímicas y Farmacéuticas, Universidad Nacional de Rosario, Suipacha 531, Rosario S2002LRK, Santa Fe, Argentina; 5Structure et Fonction des Membranes Biologiques (SFMB) Campus de la Plaine, CP206/02, Boulevard du Triomphe, 1050 Bruxelles, Belgium

**Keywords:** helix stabilization, pH sensor, coulomb interactions

## Abstract

The two-component system DesK-DesR regulates the synthesis of unsaturated fatty acids in the soil bacteria *Bacillus subtilis*. This system is activated at low temperature and maintains membrane lipid fluidity upon temperature variations. Here, we found that DesK—the transmembrane histidine kinase—also responds to pH and studied the mechanism of pH sensing. We propose that a helix linking the transmembrane region with the cytoplasmic catalytic domain is involved in pH sensing. This helix contains several glutamate, lysine, and arginine residues At neutral pH, the linker forms an alpha helix that is stabilized by hydrogen bonds in the i, i + 4 register and thus favors the kinase state. At low pH, protonation of glutamate residues breaks salt bridges, which results in helix destabilization and interruption of signaling. This mechanism inhibits unsaturated fatty acid synthesis and rigidifies the membrane when *Bacillus* grows in acidic conditions.

## 1. Introduction

Changes in environmental conditions, like temperature, osmolality, or pH, represent challenges that not only bacteria, but also multicellular organisms, have to face. Several systems have emerged in order to detect and adapt organisms to changing conditions. Nevertheless, the molecular bases of detection and signaling are still obscure for most of the systems.

Temperature changes are detected by thermosensors, which play a crucial role in poikilothermic organisms. The soil bacteria *Bacillus subtilis* is endowed with the Des pathway, which maintains lipid homeostasis at different conditions. The sensor of the pathway is the transmembrane histidine kinase DesK, which becomes activated at temperatures below 27 °C. It has been demonstrated that the transmembrane domain of DesK senses membrane fluidity changes that occur when temperature decreases. At low temperatures, the plasma membrane becomes more rigid and thicker, triggering an approach of the transmembrane segments (TMS) that generates a relative rotation in the cytoplasmic domain, allowing exposition of the catalytic His188 [1,2]. When His188 is exposed, it becomes autophosphorylated and then it transfers the phosphoryl group to the response regulator, DesR [3,4]. Phosphorylated DesR binds to the desaturase promoter in order to activate its transcription. The desaturase catalyzes the introduction of double bonds into membrane lipids in order to increase the levels of unsaturated fatty acids (UFAs) and recover lipid fluidity [5]. At high temperatures, DesK functions as a phosphatase: it removes the phosphate group from DesR; therefore, the desaturase transcription is terminated and UFA synthesis decreases (Figure 1A, [6]).

Since *Bacillus* lives in the soil, it is frequently exposed to changing environments with variations in temperature but also variations in pH. The pH of the soil varies in the range from 5.5 to 9 depending on the presence of acidic metabolites derived from organisms, chemicals derived from human activities, as well as the presence of calcite (CaCO_3_), or gibbsite (Al(OH)_3_) [7,8].

Here, we tested the biological activity of the thermosensor DesK in the pH range from 5.5 to 9. We found that DesK is inactive at low pH regardless of temperature. This raises the question: how can DesK disregard the cold signal in acidic conditions? We propose that the linker, which links the transmembrane region with the cytoplasmic catalytic domain, functions as a pH sensor and it is able to inactivate DesK at low pH. This suggests that the pH effect overcomes the well-known temperature effect on DesK activation.

## 2. Materials and Methods

### 2.1. Strains and Activity Measurements

For beta-galactosidase measurements, the *B. subtilis* JH642-CM21 cells complemented with plasmids containing DesK variants were grown in Spizizen salts (K_2_HPO_4_, 14.8 g/L, KH_2_PO_4_, 5.4 g/L, (NH_4_)_2_SO_4_, 2 g/L, tri-Sodium citrate·2H_2_O 1.9 g/L, and MgSO_4_·7H2O 0.2 g/L), adjusted to different pH with NaOH or HCl and supplemented with glycerol 0.1%, tryptophan 50 μg/mL, phenylalanine 50 μg/mL, Casamino Acids 0.05%, Lincomycin 25%, Erythromycin 0.1%, Chloramphenicol 5 μg/mL, Kanamycin 5 μg/mL, and Spectinomycin 100 μg/mL. All reagents were purchased from Sigma Aldrich with molecular biology purity. β-galactosidase activity was assayed as previously described [9,10]. The results shown are the average of five independent experiments. Standard deviations are shown with error bars.

### 2.2. Statistics

For each strain, we compared the β-galactosidase activity measured in Miller units (MU) and performed the ANOVA statistical test to determine whether the differences between the different measurements were statistically significant. In all cases, a non-parametric test was applied with R commander software. In addition, the corresponding comparison tests were performed to determine whether the MU (dependent variable) was significantly different between the different categories of the independent variable for each strain considering a *p* value less than 0.05.

### 2.3. Fluorescence Anisotropy

Total lipids from *Bacillus* cells grown at pH 6 or pH 7.5 were extracted and used to generate unilamellar liposomes [11]. Liposomes were incubated with 0.2 μg/mL DPH (1,6-diphenyl 1,3,5-hexatriene, purchased from Sigma Aldrich, Inc., Brussels, Belgium) for 15 min at 45 °C to allow homogeneous integration of the probe in the bilayer. Fluorescence anisotropy of DPH (defined as the ratio of polarized components to the total intensity by the equation r = I_II_ − I_⊥_/(I_II_ + 2I_⊥_) was used to monitor changes in membrane dynamics and was measured on a Varian Cary Eclipse fluorometer with the following protocol: time-based polarization, digital mode, excitation wavelength of 358 nm, and emission wavelength of 430 nm. Polarization was assessed using a temperature gradient from 22 to 40 °C with a 2 °C increase. Fluorescence anisotropy of non-DPH-labeled control bilayers was used for correction.

### 2.4. Synthetic Peptides 

Synthetic peptides were purchased from GL Biochem Ltd (Shangai-China). Peptides corresponding to the linker sequence KSRKERERLEEK (wild-type) and KSRKERERWEEK (with tryptophan at position 40) were N-terminal–acetylated, C-terminal–amidated, and HPLC-purified. Their mass was confirmed by mass spectrometry. For FTIR experiments, samples were diluted in 10 mM HCl and lyophilized (three times) to remove traces of TFA.

### 2.5. ATR-FTIR Measurement

Peptides were deposited on a 1-mm-square diamond ATR element fitted to a < Brucker IFS-55FTIR spectrometer. The samples (1 μL) were partially dried under nitrogen flux. Serial measurements were recorded at a 2 cm^−1^ resolution and averaged in the (4000–900) cm^−1^ spectral region with a subtracted background. Processing of the spectra (scaling, integration, and vapor subtraction) was made with in-house-made software (Kinetics) running under Matlab. Evolution of the percentage of hydrogen exchange of the peptide amide group was evaluated as a function of deuteration time. Peptides were flushed continuously with D_2_O-saturated nitrogen on the ATR element. Spectra were recorded every minute (32 scans averaged). The area of the amide II band (around 1545 cm^−1^), which is sensitive to the deuteration of the amide N-H group, was monitored over time.

### 2.6. Förster Resonance Energy Transfer (FRET) Measurement

The W-labeled peptide KSRKERERW_40_EEK was incubated at different pH with liposomes made of 75% DOPG (1,2-dioleoyl-sn-glycero-3-phospho-(1’-rac-glycerol) and 15% DOPE (1,2-dioleoyl-sn-glycero-3-phosphoethanolamine), a mixture of zwitterionic and anionic phospholipids, and labeled with dansyl-PE (10%). These reagents were purchased from Avanti Polar Lipids and have a purity higher than 99%. The proximity of the linker peptide to the membrane was studied by Förster resonance energy transfer (FRET). The intensity of the FRET is measured from the maximum emission peak of the dansyl at 515 nm and the Tryptophan emission peak between 335–350 nm. The electrostatic interaction was also checked by the increasing concentrations of KCl, purchased from Sigma Aldrich.

## 3. Results

### 3.1. DesK Senses pH

*Bacillus subtilis* cells were cultivated in a synthetic medium at different pH levels in the range from 5.5 to 9. Cells were not able to grow at pH 5.5 (Appendix A); so, DesK kinase activity was measured in the range of pH 6–pH 9 [9,10]. Strain CM21 is DesK- and contains a single-copy transcriptional fusion of the reporter gene β-galactosidase to the desaturase promoter in the amyE chromosomal locus. This strain was complemented with the plasmid pHPKS, containing a copy of the DesK gene (wild-type DesK or the truncated version, DesKC), allowing the measurement of the in vivo activity of DesK variants. Figure 1A shows that the activity is inhibited below pH 7. On the contrary, the truncated version, DesKC, which lacks the transmembrane domain, is not pH-dependent (Figure 1B). These results suggest that the protein must be anchored or integrated in the plasma membrane to detect pH changes.

We wanted to evaluate whether acidic pH decreases the levels of UFAs, which are the final product of the DesK-DesR pathway. For this, *Bacillus* cells were grown at 25 and 37 °C and at pH 6 and pH 7.5. Lipids were extracted and fatty acid composition was analyzed by gas chromatography. Figure 2A shows that the levels of UFAs are lower at pH 6 than at pH 7.5, which matches with the decrease in activity measured in Figure 1A. Cells grown at low temperature (25 °C) in acidic conditions (pH 6) (black bars, Figure 2A) synthesize lower levels of UFAs than cells grown at neutral pH. Surprisingly, UFA levels of cells grown at 25 °C (stimulating condition) and acidic conditions (inhibitory) are lower than UFA levels of cells grown at 37 °C at neutral pH (grey bars, Figure 2B), suggesting that acidic conditions hamper activation of the sensor by low temperature, the reported stimulus.

To evaluate whether the inhibition of the DesK-DesR pathway—that occurs in acidic conditions and low temperatures (25 °C), decreasing UFA levels—leads to an increase in the rigidity of the membrane, we carried out the following experiment: *Bacillus* cells were grown at 25 °C at different pH levels up to OD = 0.6, and their lipids were extracted using the Bligh and Dyer protocol. These lipids were used to prepare unilamellar liposomes by extrusion. These liposomes were incubated with the diphenylhexatriene (DPH) fluorescent probe and used to perform fluorescence polarization assays and measure the micro viscosity of the bilayer [12]. In Figure 2C, we can observe an increase in the anisotropy value in the lipids that come from cells grown at pH 6, which suggests that the lipid core of the membrane is more rigid when *Bacillus* cells grow in acidic conditions. Taken together, these results imply that there would be no compensatory mechanism to restore fluidity when the DesK-DesR system is turned off in acidic conditions.

### 3.2. Searching for the pH Sensing Motif

MS-DesK is an engineered minimized DesK variant that has been designed by Cybulski et al. [13]. It has only one chimerical transmembrane segment (instead of the original five) and can activate UFA synthesis at low temperature. To analyze if MS-DesK is sensitive to pH, strain CM21 (DesK- and containing a transcriptional fusion of the reporter gene β-galactosidase to the desaturase promoter) was complemented with the plasmid pHPKS containing the MS-DesK gene. The resulting strain was grown in acidic conditions and the activity of the reporter gene and the levels of UFAs were measured. Figure 3 and Appendix A indicate that the activity of MS is inhibited in acidic conditions, suggesting that the MS variant contains the region or motif responsible for pH sensing. Given that the truncated cytoplasmic domain is not sensitive to pH variations (Figure 1B), the pH sensor motif cannot reside in this domain, and must be located either in the extracellular tail (N-terminus), in the transmembrane helix, or in the linker region connecting the transmembrane with the cytoplasmic domain of MS. The transmembrane region of MS does not contain pH-sensitive residues in the pH range of our study; nevertheless, both the extracellular tail and the linker region contain residues whose pKa are close to physiological pH and therefore are good candidates to be pH sensors (Figure 3C).

The extracellular tail contains a histidine residue (H5, highlighted in pink in Figure 3C), which is protonated (neutral charged) at pH 6 and deprotonated (positively charged) at pH 7.5. To test whether this residue is the key to pH sensing, it was replaced by a lysine, another basic residue, giving rise to the variant H5K. Since lysine has a pKa around 11, its protonation state will not change in our pH assays, maintaining its positive charge both at pH 6 and pH 7.5. If protonation of H5 were involved in sensing, mutant strain H5K would be insensitive to pH. Figure 3A shows that the activity of the H5K mutant decreases at low pH, meaning that it is sensitive, and suggests that histidine 5 is not the main residue involved in pH detection.

On the other hand, the linker contains several glutamate residues, whose pKa is close to the physiological pH, converting this region into a good candidate to behave as a pH-dependent switching device [14]. Considering an alpha-helix contains 3.6 residues per turn, the distribution of charged residues that can give rise to a salt bridge is critical for helix stabilization [15]. In particular, glutamate and aspartate residues allow the formation of salt bridges with basic residues, like lysine or arginine, when they locate in the same face of the helix (in the register i, i + 3/4). Therefore, we evaluated the ability to respond to the pH of several MS-DesK constructions containing mutations in charged residues of the linker (Table 1).

Linker variant A3 has three positively charged residues (one lysine and two arginine) replaced to alanine; nevertheless coulombic interactions (i, i + 4 and j, j + 4) are maintained. Linker mutant Q3 has three glutamate residues replaced to glutamine, so coulombic interactions (i, i + 4 and j, j + 4) are disrupted. Figure 3A shows that the ability to respond to pH is maintained in variant A3 but disrupted in variant Q3 (Table 1, Figure 3A), suggesting that the pH-sensitivity of DesK could be related to the linker ability to alternate between two possible conformations depending on glutamate protonation: at neutral pH, glutamate residues E36 and E38 are negatively charged, and interact with lysine 32 and arginine 34 in the register i, i + 4 and j, j + 4. These intrahelical coulombic interactions stabilize the helix. At low pH, glutamate protonation breaks coulombic interactions, which results in helix instability. We propose that this dual behavior of the linker (stable vs. unstable) is key to pH sensing.

To test this idea, we designed two mutants in which salt bridge formation within the linker is either hampered or conserved by exchanging the position of glutamate and lysine residues. In the K32E/E36K linker variant, we changed the position of charged residues while keeping the distances that allow the formation of salt bridges (Linker Bridge+ or LB+). LB+ shows higher activity at higher pH and maintains pH regulation, suggesting that the particular position of charged amino acids is not important for pH sensing as long as the salt bridge is formed, stabilizing the helical conformation (Figure 3A). On the other hand, the linker mutant R34E/ E36K/ R37E hampers the formation of both salt bridges, both in the register j + 3 or j + 4 (Linker Bridge- or LB-). LB- is inactive and insensitive to pH, suggesting that coulomb interactions within the linker are needed for activity (Figure 3A, Table 1). In summary, the presence of negative and positive residues in the register i, i + 4 along the linker sequence allows the formation of helix-stabilizing salt bridges at a neutral pH, which seems to be required for pH sensing.

### 3.3. Low pH Triggers Linker Structural Changes That Turn off Kinase Activity

FTIR spectroscopy provides a key tool to evaluate secondary and tertiary protein structural alterations [16,17]. Replacement of hydrogen by deuterium in the peptide bond is extremely sensitive to environmental variations such as pH, and the kinetics of exchange can be used to detect conformational changes. We designed a peptide with the sequence of the linker in order to study if structural changes occur in this region when pH changes. Hydrogen/Deuterium exchange was measured at different pH (pH 3, 4, 6, 7, and 8) levels. A major decrease of the amide II area (1545 cm^−1^) is detected between pH 4 and 6. This reflects a drastic increase of the peptide accessibility to the aqueous environment and the unfolding of the structure (Figure 4A). Concomitantly, a shoulder at 1710 cm^−1^ is the signature of the glutamate protonation. Protonation of acidic residues result in charge neutralization and the breaking of salt bridges and favors a randomization of the structure.

The FTIR study shows that by lowering the pH, the peptide unfolds; however, this unfolding is more pronounced at pH ~4, not at pH 6, when DesK is inhibited in vivo. To explain this discrepancy and to test whether the pKa values of linker acidic residues change in the proximity of a lipid bilayer, in silico assays were carried out. Molecular dynamics simulations of the linker were run in two conditions: one in the bulk solution and the other on the surface of a lipid bilayer, which mimics *Bacillus* lipid composition [18,19]. The pKa values of glutamate residues were calculated in both systems through constant pH molecular dynamics (Table 2, Appendix A), showing a pKa shift towards higher values when the linker is close to the negatively charged lipid bilayer. Although the negative charge of the membrane surface is in part countered by the cations of the system, the strength of this negative field generates a negative electromagnetic environment, which could destabilize the deprotonated state of glutamates. Stabilization of the protonated form increases the pKa, which could be a mechanism by which glutamate residues get protonated at a higher pH. We propose that glutamate protonation at pH ~6 leads to helix destabilization and inhibition of signaling.

A Förster resonance energy transfer (FRET) assay was carried out to characterize the interaction between the linker and lipid membranes. We designed a peptide that corresponds to the linker region, but in which L40 has been replaced by a tryptophan. As a control, the same replacement was introduced in the protein. Mutant L40 W is active and responds to low temperatures (Appendix A). Liposomes labeled with the fluorescence acceptor dansyl-PE were incubated at different pH levels with the peptide L40 W. Figure 4B,C shows a higher FRET and a lower tryptophan peak intensity at an acidic pH with a transition around pH 5–6. This transition points a drastic change in the distance between the linker and the lipid interphase and shows the proximity of tryptophan to the lipid environment.

The linker-liposome interaction was substantially attenuated at increasing salt concentrations (Figure 4D), confirming the existence of an electrostatic interaction between the linker and the lipid membrane.

## 4. Discussion

Soil bacteria are frequently exposed to changing environments with variations not only in temperature but also in pH. The DesK-DesR pathway has been shown to be key to adapting *Bacillus* membranes at low temperatures [20]. The mechanisms reported so far, which have evolved to maintain pH homeostasis, are based on the consumption or expulsion of protons. For example, the F1F0 ATPase proton pump expels protons at the expense of ATP consumption, and several intracellular amino acid decarboxylases that are activated at low pH consume protons during catalysis [21]. These known mechanisms were designed to decrease intracellular proton levels; however, so far, mechanisms that restrict membrane lipid fluidity in response to acidic condition have not been identified.

Here, we found that the activity of DesK, a transmembrane kinase that modulates the levels of unsaturated fatty acids, is tightly controlled by pH. This indicates that DesK is a pH sensor, which adjusts the levels of unsaturated fatty acids according to the extracellular pH. DesK inhibition at low pH results in a decrease of membrane lipid fluidity that may restrict the entrance of substrates. Here, we propose that DesK activity is modulated by pH and identify the linker as the pH sensing motif.

At physiological pH, the positively charged residues of the linker interact with negatively charged glutamate residues and contribute to the generation of a continuous helix from the membrane to the cytoplasm. This helical conformation would favor the kinase-on conformation. At lower pH, linker glutamates get protonated, breaking salt bridges. The resulting destabilized and positively charged linker can interact with the negatively charged membrane to stabilize the unfolded form. We propose that glutamate protonation/deprotonation at the water-lipid interphase may constitute the driving force for the structural reorganization of the linker, which would be transmitted to the catalytic domain to control activity in a pH-dependent fashion.

In vivo data shows that DesK activity is inhibited at pH 6; nevertheless, the pKa of glutamate is 4.3. Molecular dynamics simulations suggest that glutamate residues located close to the water-lipid interphase show a higher pKa as compared to glutamates in bulk solution. This is probably because the electronegative field, generated by negatively charged membrane lipids, destabilizes the deprotonated form of glutamate, shifting the population to a protonated state.

In addition, glutamate protonation may be favored by the fact that protons, due to their positive charge, accumulate at the negative surface of the lipid membrane and decrease the local pH [22]. *Bacillus* membranes contain a high percentage of anionic lipids (44% phosphatidyl-glycerol and 12% cardiolipin, a diphosphatidyl-glycerol which contributes with two negative charges) [19], conferring a net negative charge to the lipid surface, which attracts protons and lowers the local pH.

We propose that the protonation/deprotonation of acidic residues involved in a salt bridge, which is located near the water-lipid interphase, can be associated with alternative signaling states of DesK. Interestingly, a cluster of positive and negative residues located at the water–lipid interface is also present in different transmembrane proteins, such as the tandem pore K channel (Kcnk1), the transient receptor potential proteins (TRPA1 isoform 2), and the mechanosensitive channel MscL (Table 3), proteins that may be subjected to pH modulation using a mechanism similar to DesK.

## 5. Conclusions

Our results suggest that *Bacillus* has developed a mechanism to restr ict the fluidity of its membranes at low pH, which could be a strategy to avoid the influx of substrates and limit growth in adverse conditions. The ability of DesK to adjust its activity according to both temperature and pH illustrates how evolution has tuned up the DesK system to be activated only when it is absolutely required and not detrimental for cell survival. This capability confers *B. subtilis*’ robustness in harsh conditions, as required for their natural adaptation but also of great interest to engineers in the metabolic and biomedical fields [23].

## Figures and Tables

**Figure 1 biomolecules-10-01183-f001:**
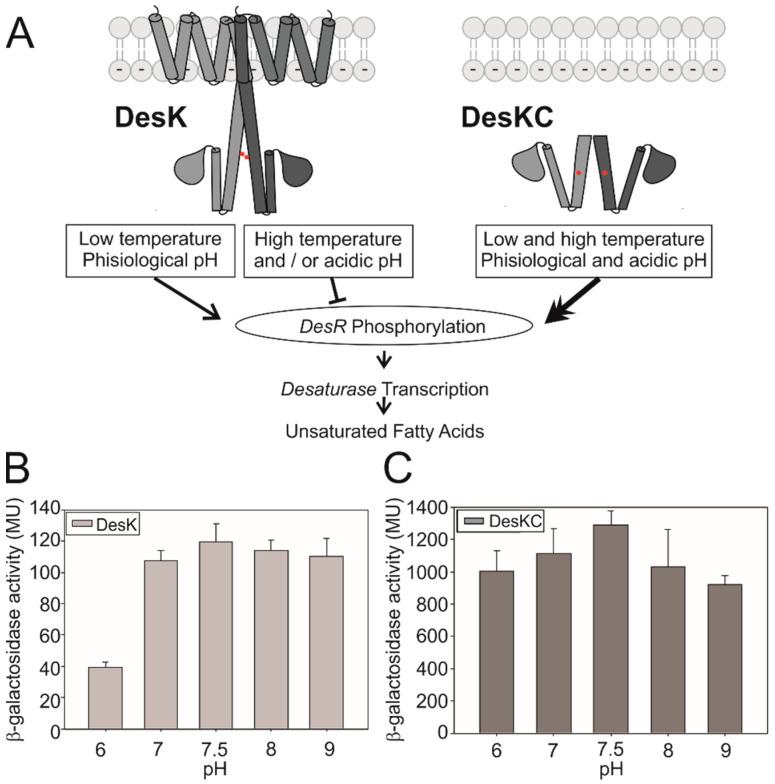
(**A**) Signal transduction of the Des pathway. A DesK dimer phosphorylates the response regulator when temperature decreases at physiological-basic pH. At high temperatures (37 °C) or with an acidic pH (pH = 6), the protein is inactive. DesKC, a truncated variant lacking the transmembrane domain, activates the system regardless of temperature or pH. DesK monomers are colored in different grey tones. The catalytic domain containing the ATP binding domain is represented by a drop and the cytoplasmic coiled-coil by a cylinder, with the phosphorylate-able histidine by a red dot. *Bacillus* strain CM21, complemented with plasmids containing the gene coding for wild-type DesK (**B**) or the truncated version, DesKC (**C**), were grown at different pH levels at 25 °C. The activity of the reporter gene β-galactosidase was determined every hour. The results shown here correspond to 4 hours of growth at the indicated pH. The activity of DesK is significantly different between pH 6 and pH 7.5, while there are not significant differences for DesKC activity at any pH. Error bars include the standard deviations from at least five independent experiments.

**Figure 2 biomolecules-10-01183-f002:**
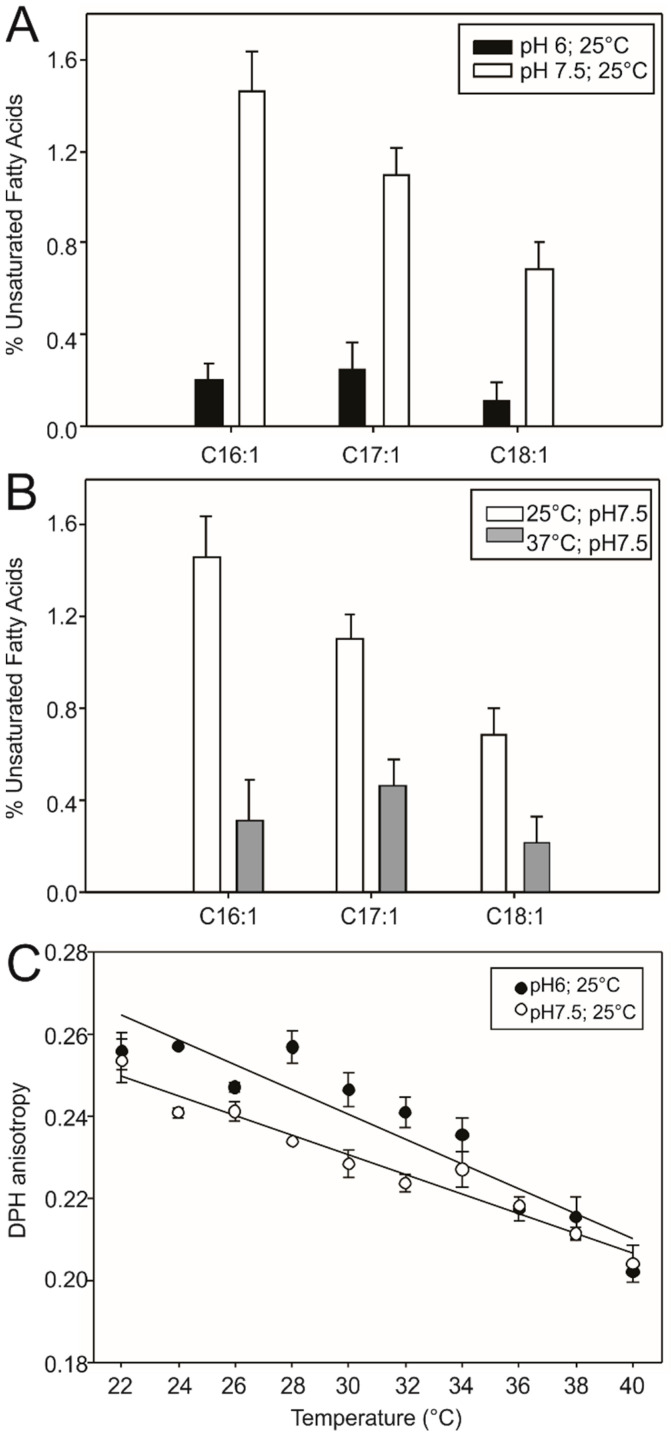
*Bacillus* lipid bilayer is less fluid in acidic conditions. (**A**) *Bacillus* cells were grown at a constant temperature (25 °C) but different pH levels (**A**); or at a constant pH (7.5) but a different temperature. (**B**) The unsaturated fatty acid (UFA) content was analyzed by gas chromatography. (**C**) *Bacillus* cells grown at 25 °C at different pH levels, and lipids extracted by Bligh and Dyer. These lipids were used to prepare liposomes that were labeled with diphenylhexatriene (DPH). Fluorescence anisotropy of DPH was used to monitor changes in membrane dynamics and was measured on a Varian Cary Eclipse fluorometer. Error bars include the standard deviations from at least five independent experiments.

**Figure 3 biomolecules-10-01183-f003:**
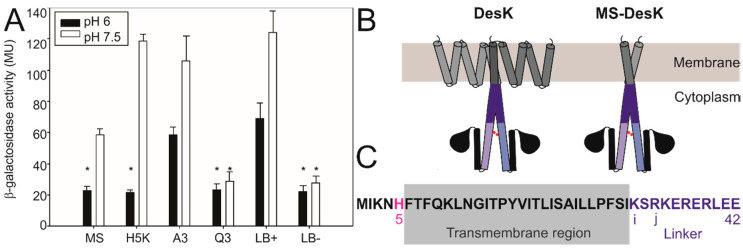
Coulomb interactions in the linker are required for pH sensing. (**A**) CM21 *Bacillus* strains expressing different MS-DesK variants were grown at 25 °C and different pH levels. The activity of the reporter gene was analyzed as in Figure 1. Error bars include the standard deviations from at least five independent experiments. There are no significant differences among the kinase activities pointed out with asterisks (*). (**B**) Drawings show a dimer of DesK and the minimized construction of MS-DesKC. The transmembrane region is represented by grey cylinders, the linker by blue cylinders, and the phosphorylate-able histidine by a red dot. (**C**) Sequence of the N-terminus, transmembrane segment, and linker of MS-DesK. The H5 is highlighted in pink, the linker motif in purple. The grey rectangle contains the transmembrane region, according to the bioinformatic software SOSUI and the curated database Uniprot (ID code: O34757).

**Figure 4 biomolecules-10-01183-f004:**
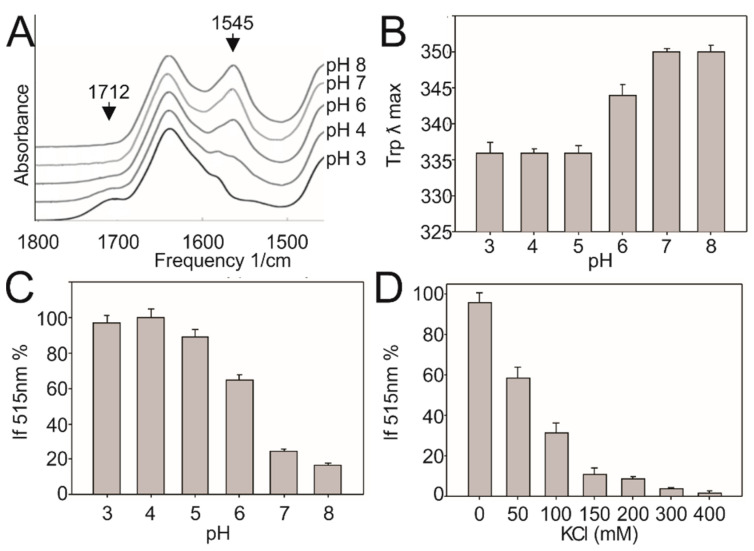
(**A**) Hydrogen/Deuterium exchange in the linker at different pH levels. Amide II area (1545 cm^−1^) is representative of H/D exchange. The accessibility to deuterium is strongly enhanced at low pH (top to bottom: pH 8, 7, 6, 4, and 3), showing there are more structures able to exchange its hydrogen by deuterium as the glutamates get protonated. The appearance of a small shoulder at 1712 cm^−1^ is indicative of glutamate protonation. (**B**) The linker peptide was incubated with liposomes made of 75% DOPG—15% DOPE and labeled with dansyl-PE. Förster resonance energy transfer (FRET) was used to study the proximity of the linker to the membrane. The Tryptophan emission peak shifts towards lower wavelengths at acidic pH levels, which indicates Trp moves to less polar environments. (**C**) The intensity of the FRET (shown in %) is measured from the maximum emission peak of dansyl at 515 nm. (**D**) Peptide-liposome samples incubated at pH 5.0 were subjected to different salt concentrations. Salt increase results in lower energy transfer, confirming the electrostatic nature of the interaction. Error bars include the standard deviations from at least five independent experiments.

**Table 1 biomolecules-10-01183-t001:** Sequence of DesK linker variants.

Linker variant	i 32		j 34		i + 4 36		j + 4 38			41	42
WT	K	S	R	K	E	R	E	R	L	E	E
A3	K	S	R	A	E	A	E	A	L	E	E
Q3	K	S	R	K	Q	R	Q	R	L	Q	E
Linker bridge+	E	S	R	K	K	R	E	R	L	E	E
Linker bridge−	K	S	E	K	K	E	E	R	L	E	E

**Table 2 biomolecules-10-01183-t002:** Glutamate pKas in each Molecular Dynamics system.

Residue	pKa in Bulk Solution	pKa Close to the Membrane
Glu 36	3.64	5.43
Glu 38	3.89	5.83
Glu 41	3.57	5.59
Glu 42	4.18	5.84

**Table 3 biomolecules-10-01183-t003:** Cluster of charged residues at the junction between transmembrane and cytosolic domains in signaling proteins and channels. Underlined residues can form salt bridges in the register i; i + 3 or i; i + 4.

Protein	i		j	i + 3	i + 4		j + 4			Organism	Uniprot ID Code
DesK	K	S	R	K	E	R	E	R	L	*Bacillus subtilis*	O34757
BvgS	K	R	A	E	R	A	L	N	D	*Bacillus pertussis*	P16575
Kcnk1	E	E	Q	K	Q	S	E	P	F	Mouse	O08581
MscL	R	K	K	E	E	P	A	A	A	*Escherichia coli*	P0A742
TRPA1.2	R	F	K	K	E	Q	M	E	Q	Chimpanzee	H2R465
Osm9	E	L	W	R	A	Q	V	V	A	Rabbit	Q9XSM3

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
