# Peer review of "A Transmembrane Histidine Kinase Functions as a pH Sensor"

_biomolecules, 2020, doi:10.3390/biom10081183_

Round 1
Reviewer 1 Report
In this work Bortolotti and co-authors extend their previous work on DesK-DesR two component system which is known to regulate synthesis of unsaturated fatty acids in Bacillus subtilis in response to temperature changes to maintain membrane fluidity. Specifically, authors show that DesK histidine kinase can also respond to pH changes. In this work it is proposed that protonation of glutamate residues at low pH could inhibit kinase activation preventing synthesis of unsaturated fatty acid. Generally, it is well executed study, however certain questions need to be addressed before publication of this work.
- Is the reporter transcriptional fusion in CM21 on the chromosome or on a plasmid? The experiments must be carried out in physiologically relevant conditions with single-copy reporter transcriptional fusion.
- How stable is MS-DesK variant as compared to full length protein? Such purified proteins should be shown and stability measured. Similarly, variants with replacement of His residues with Lys or replacement of glutamate residues must be analyzed at biochemical level to show that protein conformation is not affected.
- Authors use plasmid “encoded” gene in the results as well Material and Methods section at several places. Plasmid do not code for genes. Revise the text appropriately in all places.
- Extensive editing is needed at time pH is written as ph line 190, line 153 Bligh is mentioned as Blight line 153.
Author Response
The answers are in bold letters.
Comments and Suggestions for Authors
In this work Bortolotti and co-authors extend their previous work on DesK-DesR two component system which is known to regulate synthesis of unsaturated fatty acids in Bacillus subtilis in response to temperature changes to maintain membrane fluidity. Specifically, authors show that DesK histidine kinase can also respond to pH changes. In this work it is proposed that protonation of glutamate residues at low pH could inhibit kinase activation preventing synthesis of unsaturated fatty acid. Generally, it is well executed study, however certain questions need to be addressed before publication of this work.
- Is the reporter transcriptional fusion in CM21 on the chromosome or on a plasmid? The experiments must be carried out in physiologically relevant conditions with single-copy reporter transcriptional fusion.
The transcriptional fusion Pdes-lacZ is on the amyE locus in the chromosome; so, the experiments were carried out in physiologically relevant conditions with a single-copy transcriptional fusion. This clarification was included in page 3, lines 117-118.
- How stable is MS-DesK variant as compared to full length protein? Such purified proteins should be shown and stability measured. Similarly, variants with replacement of His residues with Lys or replacement of glutamate residues must be analyzed at biochemical level to show that protein conformation is not affected.
We agree with the reviewer that it would be nice to analyze the stability and biochemical behavior of different mutants to show that protein conformation is not affected. Nevertheless, our major objective in the present study was to focus on the biological activity of proteins and mutants in a cellular environment. Resolving the protein conformation in such environment remains one of the great challenge in structural biology and is not possible at the present time. However reconstituting DesK protein and mutants in proteoliposomes and studying how their structure depend on the lipid environment is our next challenge, but is beyond the objectives of the present study.
- Authors use plasmid “encoded” gene in the results as well Material and Methods section at several places. Plasmid do not code for genes. Revise the text appropriately in all places.
Sentences was modified in page 2 line 64, page 3 line 119, page 4 line 119 and page 6 line 173.
- Extensive editing is needed at time pH is written as ph line 190, line 153 Bligh is mentioned as Blight line 153.
Corrections were introduced.
Reviewer 2 Report
Interesting experimental paper describing a new pH-dependent response of the Des kinase system, including a mechanism, and a rationalization for its function. The paper is clearly written, easy to understand, and experimental conditions are described in detail. It integrates simulations and experiments well. For future work, the authors might want to consider adding Nuclear Magnetic Resonance work, to monitor the bilayer rigidity, peptide dynamics and to measure amide proton exchange.
Some minor changes to spelling/notation
Line 38. The soil bacteria, Bacillus
Remove the comma
Lines 42-45. Using the word “movement” twice here is ambiguous, it can be interpreted as the local dynamics being increased, while the mechanism described is a conformational change based on hydrophobic mismatch.
Figure 1. in (A) in the second box “High temperature Acidic pH” should be “High temperature and/or acidic pH” to correspond to the legend.
Throughout the manuscript a “European” decimal notation is mixed with the standard notation. Please substitute commas with periods where appropriate. A few instances are described below.
Line 56 5,5->5.5
Figure 1. legend 7,5-> 7.5
Line 142 & line 187 7,5->7.5
Figure 2. On axes and legend boxes. Please substitute commas in decimal number notation with period.
Figure 3 legend box. Please substitute commas in decimal number notation with period.
Table 2. Please substitute commas in decimal number notation with period.
Figure S1, S2. Please substitute commas in decimal number notation with period.
Etc, etc….Use coherent decimal number notation throughout.
Line 145 synthetize (ok but uncommon)-> synthesize
Line 254 extra spaces or missing spaces for pH<space>~4, pH6.
Author Response
The responses to the reviewer´s commetns are in bold letters
Comments and Suggestions for Authors
Interesting experimental paper describing a new pH-dependent response of the Des kinase system, including a mechanism, and a rationalization for its function. The paper is clearly written, easy to understand, and experimental conditions are described in detail. It integrates simulations and experiments well. For future work, the authors might want to consider adding Nuclear Magnetic Resonance work, to monitor the bilayer rigidity, peptide dynamics and to measure amide proton exchange.
Some minor changes to spelling/notation
Line 38. The soil bacteria, Bacillus
Remove the comma
The sentence was corrected in line 38.
Lines 42-45. Using the word “movement” twice here is ambiguous, it can be interpreted as the local dynamics being increased, while the mechanism described is a conformational change based on hydrophobic mismatch.
The paragraph was corrected. The change can be seen in line 43.
Figure 1. in (A) in the second box “High temperature Acidic pH” should be “High temperature and/or acidic pH” to correspond to the legend.
Second box of figure 1 was corrected regarding the suggestion.
Throughout the manuscript a “European” decimal notation is mixed with the standard notation. Please substitute commas with periods where appropriate. A few instances are described below.
Line 56 5,5->5.5
Figure 1. legend 7,5-> 7.5
Line 142 & line 187 7,5->7.5
Figure 2. On axes and legend boxes. Please substitute commas in decimal number notation with period.
Figure 3 legend box. Please substitute commas in decimal number notation with period.
Table 2. Please substitute commas in decimal number notation with period.
Figure S1, S2. Please substitute commas in decimal number notation with period.
Etc, etc….Use coherent decimal number notation throughout.
The European style -with periods- is used along all the manuscript.
Line 145 synthetize (ok but uncommon)-> synthesize
“synthesize” is now used in line 143.
Line 254 extra spaces or missing spaces for pH<space>~4, pH6.
It was corrected in line 252 and in other lines of the manuscript.
Round 2
Reviewer 1 Report
The manuscript has been improved
This manuscript is a resubmission of an earlier submission. The following is a list of the peer review reports and author responses from that submission.